# HIV Care Engagement Is Not Associated with COVID-19 Vaccination Hesitancy during the Initial Peak of the COVID-19 Pandemic among Black Cisgender Sexual Minority Men and Transgender Women in the N2 COVID Study

**DOI:** 10.3390/vaccines11040787

**Published:** 2023-04-03

**Authors:** Dustin T. Duncan, Su Hyun Park, Yen-Tyng Chen, Brett Dolotina, Wilder R. Worrall, Hillary Hanson, Mainza Durrell, Gustavo Arruda Franco, Stephen S. Morse, John A. Schneider

**Affiliations:** 1Department of Epidemiology, Columbia University Mailman School of Public Health, New York, NY 10032, USA; 2Saw Swee Hock School of Public Health, National University of Singapore and National University Health System, Singapore 117549, Singapore; 3Bloustein School of Planning and Public Policy, Rutgers University, New Brunswick, NJ 08901, USA; 4Chicago Center for HIV Elimination, University of Chicago, Chicago, IL 60637, USA; 5Survey Lab, University of Chicago, Chicago, IL 60637, USA; 6Department of Medicine, University of Chicago, Chicago, IL 60637, USA; 7Department of Public Health Sciences, University of Chicago, Chicago, IL 60637, USA; 8Crown Family School of Social Work, Policy and Practice, University of Chicago, Chicago, IL 60637, USA

**Keywords:** HIV care engagement, COVID-19 vaccination hesitancy, Black cisgender sexual minority men, Black transgender women

## Abstract

Background: Although there is limited literature on medication adherence (including HIV care engagement) and COVID-19 vaccine hesitancy in general populations (i.e., non-sexual or gender minority populations), even less is known about whether HIV care engagement correlates with COVID-19 vaccine hesitancy among sexual and gender minorities, especially those from intersectional backgrounds. The objective of the current study was to examine if an association exists between HIV status neutral care (i.e., current pre-exposure prophylaxis [PrEP] or antiretroviral therapy [ART] use) and COVID-19 vaccination hesitancy among Black cisgender sexual minority men and transgender women at the initial peak of the pandemic. Methods: We conducted the N2 COVID Study in Chicago from 20 April 2020 to 31 July 2020 (analytic *n* = 222), including Black cisgender sexual minority men and transgender women who were vulnerable to HIV as well as those who were living with HIV. The survey included questions regarding HIV care engagement, COVID-19 vaccination hesitancy and COVID-19 related socio-economic hardships. Multivariable associations estimated adjusted risk ratios (ARRs) using modified Poisson regressions for COVID vaccine hesitancy adjusting for baseline socio-demographic characteristics and survey assessment time period. Results: Approximately 45% of participants reported COVID-19 vaccine hesitancy. PrEP and ART use were not associated with COVID-19 vaccine hesitancy when examined separately or combined (*p* > 0.05). There were no significant multiplicative effects of COVID-19 related socio-economic hardships and HIV care engagement on COVID-19 vaccine hesitancy. Conclusions: Findings suggest no association between HIV care engagement and COVID-19 vaccine hesitancy among Black cisgender sexual minority men and transgender women at the initial peak of the pandemic. It is therefore essential that COVID-19 vaccine promotion interventions focus on all Black sexual and gender minorities regardless of HIV care engagement and COVID-19 vaccine uptake is likely related to factors other than engagement in HIV status neutral care.

## 1. Introduction

Due to the material impacts of systemic racism, homophobia, and transphobia within society and medicine [1,2,3,4], quality of health care widely varies by race, sexuality, and gender identity. Given this, it is not surprising that racial disparities in COVID-19 infection, hospitalization and mortality exist which reflect pre-existing and persistent racial disparities in opportunities and access to resources due to structural issues, including structural factors related to the COVID-19 vaccine accessibility as well as racialized medical mistrust [5]. In addition, there are sexual orientation disparities in COVID-related health outcomes including vaccine uptake [6], but less is known about these disparities among intersectional populations, especially among Black cisgender sexual minority men and Black transgender women [6]. Vaccine hesitancy, i.e., a delay in the acceptance or refusal of vaccines despite the availability of vaccination services, is a particularly under-studied topic among Black cisgender sexual minority men and Black transgender women in the COVID-19 pandemic, which is a major gap in the literature.

Prior studies have shown that members of racial minority groups are often less likely to seek out the COVID-19 vaccination [7,8]. Indeed, in one study of an online sample of sexual minority men and transgender women [9], Black participants were significantly less likely to accept a COVID-19 vaccine, compared to their White counterparts. Another study of 101 Black people living with HIV (PLWH) found that 97% of participants endorsed at least one general COVID-19 mistrust belief, with greater medical mistrust being related to greater vaccine and treatment hesitancy regarding COVID-19 [10]. Interestingly, COVID-19 conspiracy beliefs were high among a sample of Black cisgender sexual minority men and Black transgender women in Chicago; for example, three-quarters of the sample believed at least one conspiracy theory that COVID-19 was either government-created or lab-created accidentally or purposefully [11].

Several factors appear to be correlated with higher COVID-19 vaccine likelihood, including engagement in HIV care [12]. Another study of 1030 PLWH demonstrated that individuals adherent to antiretroviral therapy (ART) adherence (noted by a CD4 count of 200 cells/mm^3^) were more likely to receive the COVID-19 vaccine [8], which has also been shown in international samples of PLWH [13]. Individuals who engaged in HIV care may also be in more frequent contact with their healthcare providers and generally more health-conscious, i.e., may have greater access to care and engagement in the healthcare system and due to this and/or other factors may focus on their health more than those not engaged in the healthcare system. Regarding pre-exposure prophylaxis (PrEP), however, we are only aware of one study that examined the association between PrEP engagement and COVID-19 intention [14]. This study found no association between PrEP engagement and COVID-19 intention and also found no association between substance use treatment and hospitalization for mental health with COVID-19 vaccine hesitancy.

Although there is limited literature on HIV care engagement and COVID-19 vaccine hesitancy with mixed findings thus far, there are very few studies that have specifically focused on intersectional populations, including Black sexual and gender minority people who have endured historical racism and discrimination that negatively impact healthcare engagement [2,3]. Moreover, although a few recent studies have included mixed HIV status cohorts [11,15,16], the large majority of the literature remains stratified by HIV status [17], which may contribute to HIV stigma. The objective of the current study was to examine whether an association exists between HIV care engagement (i.e., current PrEP or ART use) and COVID-19 vaccination hesitancy among Black cisgender sexual minority men and transgender women at the initial peak of the pandemic. We hypothesized that engagement in HIV care engagement will be associated with a lower likelihood of COVID-19 vaccine hesitancy among Black sexual minority men and Black transgender women due to greater access to care and engagement in the healthcare system, which increases healthcare seeking behaviors.

## 2. Methods

### 2.1. Data

The Neighborhoods and Networks (N2) Cohort Study is an ongoing cohort study investigating the impact of neighborhood- and network-level factors on HIV prevention and care behaviors in over 600 Black cisgender sexual minority men and Black transgender women in the Midwest (Chicago, IL, USA) and the Southern U.S. (Jackson, Mississippi and New Orleans and Baton Rouge, LA, USA). The study has been previously described in detail [15,18,19]. In response to the COVID pandemic, we developed the N2 COVID Study from 20 April 2020 to 31 July 2020 in Chicago (*n* = 226). To do so, we contacted 405 of the 412 N2 baseline participants in Chicago and were able to reach 226 of them for the N2 COVID Study. Survey interviews were conducted via Zoom by highly trained interviewers at the Survey Lab at the University of Chicago. The interview time lasted forty minutes on average. At the conclusion of the interview, participants were given a $35 incentive. Referrals to social and health services (e.g., unemployment benefits and COVID-19 testing) were also provided as needed.

Participants were tested for HIV at the baseline survey. Due to social distancing protocols, we did not test participants for HIV during the COVID-19 check-in survey. Participants’ HIV status for the current study was determined using (1) the diagnostic test from the baseline data collection and (2) self-reported responses during the N2 COVID-19 check-in survey. Participants not living with HIV tested negative for HIV at baseline and self-reported HIV negative at the COVID-19 check-in survey. Participants living with HIV tested positive for HIV at baseline or self-reported HIV positive at the COVID-19 check-in survey. Comparing the 412 N2 baseline participants in Chicago and the N2 COVID Study sample in Chicago of 226, we found few (but some) socio-demographic differences (e.g., no differences in age or income but differences in housing instability and employment).

The Biological Sciences Division/University of Chicago Medical Center Institutional Review Board (IRB) at the University of Chicago has reviewed and approved all protocols to be implemented at the Chicago Center for HIV Elimination. In addition, the Columbia University Mailman School of Public Health IRB has reviewed and approved all protocols for the N2 Study.

The analytic sample included 222 participants. Four participants were dropped for testing HIV-positive at baseline but reported being HIV-negative in the subsequent N2 COVID-19 survey.

### 2.2. HIV Status Neutral Care Engagement

We measured if the frequency of PrEP use or ART use had changed since the shelter-in-place order. For participants not living with HIV, we asked if they were currently using PrEP (yes/no). If they reported currently using PrEP, we further asked, “Since the shelter-in-place order, have you used PrEP more, less, or about as often as you used it before the pandemic”? Response options included “Used PrEP more during COVID-19”, “Used PreP less during COVID-19”, or “About as often before and during pandemic”.

For participants living with HIV, we asked if they were using ART (yes/no). If they reported ART use, we further asked, “Since the shelter-in-place order, have you missed doses of your HIV medication more, about the same, or less often than before the pandemic started”? Response options included “Missed dose of HIV medication more frequently during COVID-19 pandemic”, “HIV medication use about the same before and during the pandemic”, or “Missed doses of HIV medications less frequently during the COVID-19 pandemic”.

Based on the questions above, we examined PrEP and ART use in the initial peak of COVID-19 separately. More specifically, we assessed whether someone was not on PrEP or reported using PrEP less since the pandemic for HIV-negative participants; similarly, for participants living with HIV, the participants not using ART and those reporting missing more doses of ART. Both PrEP and ART use were also grouped together due to low cell sizes otherwise [11]. That is, we created a new variable of HIV status neutral care to include whether the participate used PrEP or ART at that time.

### 2.3. COVID-19 Vaccination Hesitancy

The outcome of COVID-19 vaccination hesitancy was measured with the question: “How likely are you to get vaccinated for COVID-19 once a vaccination is available to the public?” This was evaluated with a four point Likert scale, ranging from very unlikely to very likely. For analysis, we combined the somewhat unlikely, somewhat likely, and very likely groups together to compare to the very unlikely group. As part of the exploratory analysis, we also created an outcome of combining very unlikely and somewhat unlikely compared to somewhat likely and very likely. Participants who refused or answered “Don’t know” were dropped from analysis (*n* = 6).

### 2.4. Socio-Demographic Characteristics

From the baseline survey, we included the following socio-demographic characteristics: age in years, gender identity, sexual orientation, relationship status, education (binary coded as high school or higher vs. no high school and nothing higher), being employed, annual income (binary coded as ≥$20,000 USD vs. <$20,000 USD), and housing stability (“history of housing stability”). In addition, baseline use of PrEP or ART was included in this study.

In addition, for this study, we selected two-time frames of the pandemic to disaggregate the data: 20 April 2020 to 2 June 2020 (Lockdown/Phase1/2) and 3 June 2020 to 31 July 2020 (Phase 3/post). These timelines were chosen based on “Restore Illinois”, the gubernatorial mandated public health reopening schedule for the state of Illinois, which corresponded to the different degrees of restrictions that residents of Chicago experienced during the pandemic.

### 2.5. Statistical Analyses

We first conducted descriptive statistics for the full sample of participants. After descriptive statistics were computed, bivariable and multivariable associations between HIV care engagement and COVID vaccine hesitancy (using both ways to define the outcome variable) during the peak of the COVID-19 pandemic in Chicago were performed. Multivariable associations estimated adjusted risk ratios (ARRs) using modified Poisson regressions for COVID vaccine hesitancy adjusting for baseline socio-demographic characteristics, baseline PrEP/ART use and survey assessment time period. Modified Poisson regressions were chosen as this technique robustly estimates ARRs rather than the odds ratio [20]. For statistical hypothesis testing, 95% confidence intervals (CI) were estimated as well as *p*-values. A statistical significance level of 0.05 was applied.

## 3. Results

Table 1 shows descriptive statistics on our sample of cisgender Black sexual minority men and Black transgender women. Among a total of 222 participants, the mean age was 25.76 (SD = 4.04) years, and 88.3% were cisgender male. More than a half reported being gay (58.1%) and 60.8% reported being single, almost 90% reported that they completed high school or higher education (89.6%), 62.2% reported that they had an annual income less than $20,000, and 68.0% reported that they had stable housing in the past 3 months. In addition, 59.5% of the sample were not living with HIV and 40.5% of the sample were living with HIV. Almost half of the sample (45.0%) reported not being likely to uptake the COVID-19 vaccine, whereby 32.4% reported that they were very unlikely to update the COVID-19 vaccine. Among people living with HIV (*n* = 90), 8.2% reported that they were not on ART and 15.3% reported that they missed ART more since SIP. Among people who were not living with HIV (*n* = 132), 68.2% reported that they were not on PrEP and 8.3% reported that they used PrEP less since SIP.

### Association between HIV Care Engagement and COVID-19 Vaccination Hesitancy

Table 2 shows the association between HIV care engagement and COVID-19 vaccination hesitancy (we present results of both ways to define our including as represented in Model 1 and Model 2) among our sample of cisgender Black sexual minority men and transgender women. In multivariable models (adjusting for age, gender identity, orientation, relationship status, education, annual income, employment, housing stability and survey assessment time period and baseline PrEP or ART use), no significant associations were observed between PrEP use and COVID-19 vaccine hesitancy (Model 1: ARR = 1.51, 95% CI = 0.70, 3.25; Model 2: ARR = 0.89, 95% CI = 0.55, 1.44) as well as ART use and COVID-19 vaccine hesitancy (Model 1: ARR = 1.74, 95% CI = 0.65, 4.71; Model 2: ARR = 0.68, 95% CI = 0.29, 1.58). Combined PrEP and ART use (HIV status neutral care) was also not associated with COVID-19 vaccine hesitancy (Model 1: ARR = 1.40, 95% CI = 0.88, 2.23; Model 2: ARR = 0.84, 95% CI = 0.59, 1.20) in multivariable models.

Additionally, there were no significant multiplicative effects of COVID-19 related socioeconomic hardships and HIV care engagement on COVID-19 vaccine hesitancy (results not shown). More specifically, in an exploratory analysis, we assessed interaction effects by adding multiplicative interaction terms—i.e., (1) COVID-19 related income, (2) COVID-19 related housing stability, and (3) COVID-19 related food insecurity—to the multivariable models but did not find any significant multiplicative effects (*p* > 0.05).

## 4. Discussion

Several studies have shown an association between HIV care engagement and COVID-19 vaccine hesitancy in general (i.e., non-sexual or gender minority) samples. In this study, we examined if an association exists between HIV status neutral care engagement (i.e., current PrEP or ART use) and COVID-19 vaccination hesitancy among Black cisgender sexual minority men and transgender women at the initial peak of the pandemic. We used an HIV status neutral care approach [17], combining participants who used PrEP and ART, which allowed us to examine our research question with a more robust sample size than if we only separated participants by medication type. Very few other studies have utilized this innovative public health approach [11,15,16], and our study contributes to this growing literature by employing this approach to explore HIV care engagement and COVID-19 vaccine hesitancy among Black cisgender sexual minority men and transgender women.

We found that PrEP use was not associated with COVID-19 vaccine hesitancy nor was ART use associated with COVID-19 vaccine hesitancy during the initial peak of the pandemic. HIV status neutral care engagement (combined PrEP and ART use) was also not associated with COVID-19 vaccine hesitancy. Furthermore, we also found no evidence of effect modification by COVID-19 related individual-level socio-economic status, further demonstrating the robustness of our findings. Other studies have found similar results among diverse samples of people living with HIV; indeed, one study, published in 2021, of 2740 PLWH in China found no association between HIV care engagement (i.e., received antiretroviral therapy, undetectable viral load) and willingness to receive the COVID-19 vaccine [21]. Moreover, one 2022 study of 440 people at-risk for and/or living with HIV in Los Angeles and New Orleans demonstrated that PrEP/PEP use was not associated with general vaccine attitude or COVID-19 prevention behaviors [14]. However, a different study of 8033 PLWH in Oregon found that HIV care engagement was associated with higher COVID-19 vaccine uptake [12], highlighting the mixed nature of the current literature. Further, socio-economic status played an important role in vaccine uptake among prior samples of sexual and gender minorities in New York [22], unlike the findings from our study. Taken together, our study adds a meaningful contribution to the COVID-19 vaccine hesitancy literature by conducting an HIV status neutral approach and by focusing on an intersectional population who are marginalized.

Our findings may be explained by several factors. First, Black sexual and gender minority individuals have reported discrimination in healthcare settings since the pandemic which was associated with COVID-19 vaccine uptake [23]. As such, it is possible that Black individuals may engage in HIV care out of perceived necessity (which may not be pronounced as COVID-19) while still retaining strong racialized medical mistrust of the COVID-19 vaccine and treatment, which has been shown to be associated with COVID-19 vaccine hesitancy among Black PLWH [10]. In addition, overall pandemic fatigue and vaccine fatigue are also potential issues that may explain our findings. Indeed, although these data were collected before the COVID vaccine came out, participants may be overburdened by many different vaccines and treatments, particularly Black patients living with HIV. Thus, the potential vaccine fatigue indicated in our study is not the fatigue toward COVID-19 vaccine. Furthermore, there may be specific COVID-19-related misinformation that Black sexual and gender minorities encounter. Our findings suggest that future interventions should work to cultivate trust in COVID-19 vaccines and reduce vaccine fatigue among all Black sexual and gender minority individuals, regardless of whether they are engaged in HIV care and regardless of their socio-economic status. Clinicians and advocates can develop meaningful interventions by centering an authentic community investment that recognizes these discriminatory experiences in medicine [24] as well as the reality of pandemic and vaccine fatigue that minoritized groups disproportionately experience. We note, however, that our data are from 2020, so many parameters, e.g., COVID-19 knowledge and behavior, have likely changed over the last 3 years. Given this, we suggest continuing/ongoing data collection on COVID-19 vaccine hesitancy and vaccine uptake among Black SMM and TW as the pandemic continues to evolve.

In addition, although it is possible that our results indicate that there is no true relationship between HIV care engagement and COVID-19 vaccine hesitancy, it is also possible that these findings are due to limitations with our data. First, our cross-sectional study design limits causal inference. Second, our use of self-reported data is implicated in social desirability bias and recall bias. Third, like all observational research, residual confounding is also a concern. Fourth, these data were collected at the initial peak of the pandemic. It is not clear, therefore, if these findings would be different three years later in the pandemic [25]. Finally, generalizability and limited statistical power are concerns due to the relatively small sample size, geographic-specific sample and limited number of Black transgender women. Despite these limitations, our findings are bolstered by an innovative HIV-neutral approach as well as an objective measure for HIV status at baseline.

## 5. Conclusions

Findings from our analysis suggest no association between HIV care engagement and COVID-19 vaccine hesitancy among Black cisgender sexual minority men and transgender women at the initial peak of the pandemic. COVID-19 vaccine promotion interventions should focus on all Black sexual and gender minorities regardless of HIV status neutral care engagement including perhaps relying on the house/ballroom community [26]. While our findings suggest that COVID-19 vaccine uptake is likely related to factors other than engagement in HIV status neutral care, these findings need to be replicated including more recent data in the ongoing COVID-19 pandemic.

## Figures and Tables

**Table 1 vaccines-11-00787-t001:** Descriptive statistics of the N2 COVID Study of Black cisgender sexual minority men and Black transgender women (N = 222).

	Total
**Socio-demographics**	
**Age, mean (SD)**	25.76 (4.04)
**Gender identity**	
Male	196 (88.3)
Trans feminine	20 (9.0)
Other	6 (2.7)
**Sexual orientation**	
Gay	129 (58.1)
Bisexual	61 (27.5)
Straight/other	26 (11.7)
**Relationship status**	
Single	135 (60.8)
In a relationship	84 (37.8)
**Education attainment**	
Less than high school	23 (10.4)
High school or higher	199 (89.6)
**Employed**	
No	94 (42.3)
Yes	128 (57.7)
**Annual income**	
<$20,000 USD	138 (62.2)
≥$20,000 USD	82 (36.9)
**Stable housing in the past 3 months**	
No	67 (30.2)
Yes	151 (68.0)
**HIV status**	
HIV positive	90 (40.5)
HIV negative	132 (59.5)
**Survey assessment time period**	
Phase 1 (20 April–2 June)	76 (34.2)
Phase 2 (3 June–31 July)	146 (65.8)
**HIV Status Neutral Care**	
**ART use since SIP**	
Not on ART	7 (8.2)
Missed ART less	25 (29.4)
Missed ART same amount	40 (47.1)
Missed ART more	13 (15.3)
**PrEP use since SIP**	
Not on PrEP	90 (68.2)
Used PrEP less	11 (8.3)
Used PrEP same amount	26 (19.7)
Used PrEP more	5 (3.8)
**HIV neutral care**	
Used PrEP less or missed more doses of ART	121 (54.5)
Used PrEP same/more or missed fewer/same doses of ART	96 (43.2)
**COVID-19 Vaccine Uptake Likelihood**	
Very unlikely	72 (32.4)
Somewhat unlikely	28 (12.6)
Somewhat likely	42 (18.9)
Very likely	74 (33.3)

**Table 2 vaccines-11-00787-t002:** Association between HIV status neutral care and COVID-19 vaccine hesitancy among Black cisgender sexual minority men and Black transgender women.

	Model 1 (Very Unlikely vs. Others)	Model 2 (Very/Somewhat Unlikely vs. Very/Somewhat Likely)
	ARR (95% CI)	ARR (95% CI)
**PrEP exposure**		
Did not use PrEP or used PrEP less	1.51 (0.70, 3.25)	0.89 (0.55, 1.44)
Used PrEP same/more	Ref	Ref
**ART exposure**		
Did not use ART or missed ART more	1.74 (0.65, 4.71)	0.68 (0.29, 1.58)
Missed ART same/less	Ref	Ref
**HIV status neutral care** **(Combined PrEP & ART)**		
Used PrEP less or missed more doses of ART	1.40 (0.88, 2.23)	0.84 (0.59, 1.20)
Used PrEP same/more or missed fewer/same doses of ART	Ref	Ref

Adjusted for age, gender identity, orientation, relationship status, education, annual income, employment, housing stability, survey assessment time period and baseline use of PrEP or ART.

## Data Availability

The data presented in this study are available on request from the corresponding author. The data are not publicly available due to privacy and ethical restrictions.

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
