# Peer review of "HIV Care Engagement Is Not Associated with COVID-19 Vaccination Hesitancy during the Initial Peak of the COVID-19 Pandemic among Black Cisgender Sexual Minority Men and Transgender Women in the N2 COVID Study"

_vaccines, 2023, doi:10.3390/vaccines11040787_

Round 1

Reviewer 1 Report

HIV care engagement is not associated with COVID-19 Vaccination Hesitancy During the Initial Peak of the Pandemic Among Black Cisgender Sexual Minority Men and Transgender Women in the N2 COVID Study 

Dustin T. Duncan

The relationship between HIV care engagement and COVID-19 Vaccination Hesitancy among Black Cisgender Sexual Men and Transgender Women is not well established. Furthermore, rare studies have focused on intersectional populations, including Black sexual and gender minority people who has endured historical racism and discrimination may negatively impact healthcare engagement. 

To better assess such issues, investigators used the Neighborhoods and Networks (N2) Cohort Study evaluated the impact of neighborhood- and network-level factors on HIV prevention and care behaviors in over 600 Black cisgender SMM and Black TW in Chicago, Illinois and in the Deep South (Jackson, Mississippi and New Orleans and Baton Rouge, Louisiana).

During April et July, 2020, PrEPers, former PrEPers and PLWH not using ART and using ART were selected and respond to the question “How likely are you to get vaccinated for COVID-19 once a vaccination is available to the public?

Investigators did no find association between HIV care engagement and COVID-19 vaccine hesitancy among Black cisgender sexual minority men and transgender women at the initial peak of the pandemic in USA, April to July 2020.

Study is interesting. However, many comments in the discussion are based on 2023 current understanding of COVID-19 pandemic like vaccine fatigue that developed in the late 2021 and Omicron time.

The population is heterogeneous from Chicago and also call Deep South, which is a judgemental term, should be geographic. The culture, income, historical perspective are different in these 2 part of the country. 

Results and discussion should be more descriptive with lower level of free interpretation of the data. 

Data from 2020 are late commers as so many parameters, knowledge, behavior have changed over the last 3 years, limiting the value of such data findings. 

Author Response

Reviewer #1

The relationship between HIV care engagement and COVID-19 Vaccination Hesitancy among Black Cisgender Sexual Men and Transgender Women is not well established. Furthermore, rare studies have focused on intersectional populations, including Black sexual and gender minority people who has endured historical racism and discrimination may negatively impact healthcare engagement. 

To better assess such issues, investigators used the Neighborhoods and Networks (N2) Cohort Study evaluated the impact of neighborhood- and network-level factors on HIV prevention and care behaviors in over 600 Black cisgender SMM and Black TW in Chicago, Illinois and in the Deep South (Jackson, Mississippi and New Orleans and Baton Rouge, Louisiana).

During April et July, 2020, PrEPers, former PrEPers and PLWH not using ART and using ART were selected and respond to the question “How likely are you to get vaccinated for COVID-19 once a vaccination is available to the public?

Investigators did no find association between HIV care engagement and COVID-19 vaccine hesitancy among Black cisgender sexual minority men and transgender women at the initial peak of the pandemic in USA, April to July 2020.

Response: Thank you for such a great overview of the study background and methods as well as highlighting the importance of our study.

Study is interesting. However, many comments in the discussion are based on 2023 current understanding of COVID-19 pandemic like vaccine fatigue that developed in the late 2021 and Omicron time.

Response: We appreciate the reviewer comments. We have adjusted our Discussion to reflect the pandemic near where the data was collected. While we kept some of the language around COVID-19 in 2023, we now explicitly highlight that these new developments do not directly relate to our current findings. Thank you for this suggestion.

The population is heterogeneous from Chicago and also call Deep South, which is a judgemental term, should be geographic. The culture, income, historical perspective are different in these 2 part of the country. 

Response: We have adjusted our terminology from Deep South to Southern United States. However, we note that the N2 COVID Study focused on Chicago only, which we highlight better in our revised manuscript. Thank you for the suggestion.

Results and discussion should be more descriptive with lower level of free interpretation of the data. 

Data from 2020 are late commers as so many parameters, knowledge, behavior have changed over the last 3 years, limiting the value of such data findings. 

Response: As suggested by the reviewer, we discuss the descriptive data more with less interpretation of the data. In addition, we have further highlighted the importance of our study. To our knowledge, this is the first COVID-19 vaccine hesitancy study among young black sexual minority men, which is important to know.

Reviewer 2 Report

COVID-19 vaccine hesitancy has been studied from a variety of perspectives in order to understand how to increase the percent of the population that is fully vaccinated. Research has uncovered many of the reasons for vaccine hesitancy within the general population but there are a number of unique populations that are at higher risk of serious complications from COVID that have not been studied. Your study examines one such population.  I have made several comments and suggestions that I hope you will find useful in your current and future research in this area.

Introduction: You note that there have been “…very few studies that have specifically focused on intersectional populations…..who has endured historical racism and discrimination…..” Although the issue of racism and discrimination are not specifically mentioned in the methodology, are we to assume that your study population has documented evidence of these characteristics?  If not, I see no reason to call out these issues.

Methods: Can you provide additional information regarding subjects from the original 412 N2 participants who did not participate in the final N2 COVID study. That would include 7 who were not contacted and 179 who were not “reached.”

You indicate that you looked at two time frames (April 20 – June 2 and June 3 – July 31).  Univariate statistics are not reported for this variable. Since all other variables seem to be included in Table 1, I am curious why this variable was omitted.

Discussion: You again note that this is a population that has experienced discrimination. While I think most people would agree with that assertion, I think it would be valuable to provide any data that would confirm this.

Author Response

Reviewer #2

COVID-19 vaccine hesitancy has been studied from a variety of perspectives in order to understand how to increase the percent of the population that is fully vaccinated. Research has uncovered many of the reasons for vaccine hesitancy within the general population but there are a number of unique populations that are at higher risk of serious complications from COVID that have not been studied. Your study examines one such population.  I have made several comments and suggestions that I hope you will find useful in your current and future research in this area.

Response: Thank you.

Introduction: You note that there have been “…very few studies that have specifically focused on intersectional populations…..who has endured historical racism and discrimination…..” Although the issue of racism and discrimination are not specifically mentioned in the methodology, are we to assume that your study population has documented evidence of these characteristics?  If not, I see no reason to call out these issues.

Response: Thank you for this comment. We believe our findings are important and relate to historical racism and discrimination. We added some additional references to support these statements (e.g., Arnold et al., 2014).

Arnold EA, Rebchook GM, Kegeles SM. ’Triply cursed’: racism, homophobia and HIV related  stigma are barriers to regular HIV testing, treatment adherence and disclosure among young Black gay men. Cult Health Sex 2014;16:710–22.

Methods: Can you provide additional information regarding subjects from the original 412 N2 participants who did not participate in the final N2 COVID study. That would include 7 who were not contacted and 179 who were not “reached.”

Response: In the revised manuscript, we added information comparing the 412 participants compared to 226 sample. See Table 1 below. In addition, we also detailed the participants who were not contacted.

Supplemental Table 1. Comparison of N2 Covid baseline and study subjects.

Study subjects (n=222)

Non-study subjects (n=190)

p-value

Age, mean(SD)

2.62 (1.12)

2.60 (1.04)

0.91

Sexual orientation

   Gay

129 (59.7)

110 (59.1)

0.74

   Bisexual

61 (28.2)

49 (26.3)

   Straight/other

26 (12.0)

27 (14.5)

Education attainment

    Less than high school

23 (10.4)

35 (18.4)

0.02

    High school or higher

199 (89.6)

155 (81.6)

Employed

    No

94 (42.3)

117 (61.6)

<0.0001

    Yes

128 (57.7)

73 (38.4)

Annual income

    < $20,000 USD

138 (62.2)

131 (69.0)

0.120

    ≥ $20,000 USD

83 (36.9)

56 (29.5)

Stable housing in the past 3 months

    No

67 (30.2)

75 (39.5)

0.02

    Yes

151 (68.0)

104 (54.7)

You indicate that you looked at two time frames (April 20 – June 2 and June 3 – July 31).  Univariate statistics are not reported for this variable. Since all other variables seem to be included in Table 1, I am curious why this variable was omitted.

Response: In the revised manuscript, we added information comparing the two time frames, which we agree are an important part of our study. Thank you for this important suggestion.

Discussion: You again note that this is a population that has experienced discrimination. While I think most people would agree with that assertion, I think it would be valuable to provide any data that would confirm this.

Response: In the revised manuscript, we added information on discrimination among our intersectional samples in particular Black sexual minority men and transgender women (e.g., Turpin et al., 2021; Malebranche et al., 2004; Salerno et al., 2020).

Turpin R, Khan M, Scheidell J, Feelemyer J, Hucks-Ortiz C, Abrams J, Cleland C, Mayer K, Dyer T. Estimating the Roles of Racism and Homophobia in HIV Testing Among Black Sexual Minority Men and Transgender Women With a History of Incarceration in the HPTN 061 Cohort. AIDS Educ Prev. 2021 Apr;33(2):143-157. 

Malebranche DJ, Peterson JL, Fullilove RE, Stackhouse RW. Race and sexual identity: perceptions about medical culture and healthcare among Black men who have sex with men. J Natl Med Assoc. 2004 Jan;96(1):97-107.

Salerno JP, Turpin R, Howard D, Dyer T, Aparicio EM, Boekeloo BO. Health Care Experiences of Black Transgender Women and Men Who Have Sex With Men: A Qualitative Study. J Assoc Nurses AIDS Care. 2020 Jul-Aug;31(4):466-475. 

Round 2

Reviewer 1 Report

issues have beeb addressed